# Amidated *Pectic Polysaccharides* (Pectin) as Methane Hydrate Inhibitor at Constant Cooling and Isobaric Condition

**DOI:** 10.3390/polym15092080

**Published:** 2023-04-27

**Authors:** Adam Daniel Effendi, Muhammad Aslam Md Yusof, Nor Fariza Abd Mutalib, Chee Wee Sia

**Affiliations:** 1Department of Petroleum Engineering, Universiti Teknologi PETRONAS, Seri Iskandar 32610, Malaysia; 2Department of Chemical Engineering, Universiti Teknologi PETRONAS, Seri Iskandar 32610, Malaysia

**Keywords:** amidated pectin, amide functional group, gas hydrate, constant cooling, green alternatives, isobaric condition

## Abstract

This study aims to address the environmental impact of using common commercial hydrate inhibitors such as Methanol (MeOH) in extremely cold oil and gas environments. As a greener alternative, *Pectic Polysaccharides* (pectin) can act as a kinetic hydrate inhibitor (KHI) to delay hydrate formation. We evaluated the performance of amidated pectin (AMP), a type of pectin with higher electronegative functional groups, using a high-pressure micro-differential scanning calorimeter (HP µ-DSC) under isobaric conditions with constant cooling. We compared AMP to low-methoxylated pectin (LMP) and high-methoxylated pectin (HMP) and found that AMP was the best KHI among the tested pectin types. At a concentration of 1.0 wt.%, the AMP Relative Inhibitor Performance (RIP) was 0.10, and at 0.1 wt.%, it had an RIP of 0.07, which were the only positive RIPs obtained amongst the tested KHIs. The results suggest that AMP can be a sustainable KHI option in extremely cold environments where the KHI effectiveness typically declines.

## 1. Introduction

Industrial activities in the Artic region had increased significantly, ranging from wind farms to oil and gas development. This region is often characterized as harsh and has severe climatic conditions and is environmentally sensitive [1,2,3]. Its infrastructure is less developed and far from any main industrial hubs. The complex operational environment severely limits operational options and affects equipment performance. Moreover, it became a logistical challenge when considering the possibility of the long downtime caused by the uncertain harsh weather conditions. When implementing an operating practice in this environment, one must consider the logistical limitation and the strict environmental regulation in place, such as the standard set by the Oslo/Paris convention (OSPAR) [4]. Reducing the operational scope toward the oil and gas industry, gas hydrate formation issues would be frequent in the cold north.

The gas field in the region would have the aforementioned challenges; thus, a green alternative would suit their gas hydrate prevention measure. The chemical method for hydrate formation could be divided into thermodynamic hydrate inhibitors (THIs) and low-dosage hydrate inhibitors (LDHIs), of which the LDHIs further split into kinetic hydrate inhibitors (KHIs) and Anti-Agglomerants (AA). However, due to the environmental restriction, it could make it difficult to use a common commercial hydrate inhibitor due to its toxicity or poor biodegradation [5,6]. The logistical issue also prevents the use of thermodynamic hydrate inhibitors (THIs). Hence, the use of low-dosage hydrate inhibitors (LDHIs) would be ideal as it requires a significantly lower volume of hydrate inhibitors to prevent hydrate formation [4,5,7].

The two types of LDHIs, the KHI and AA, have their own merits and disadvantages. An AA has a performance requirement where it could not be used in a gas field [8,9]. Furthermore, Lavallie et al. [10] reported successful applications of a KHI in the DOLPHIN field in Qatar which is a sour gas/condensate field and have also discussed their injection procedure. They utilized continuous injection with the first injection being performed offshore to mitigate the risk of hydrate formation, and the second injection was carried out onshore as required at the inlet [10]. Hence, a KHI would be chosen to be optimal for gas hydrate formation inhibition. The following are the summarized findings of the chemical method used in preventing hydrate formation (Table 1) based on Othman’s [11] findings:

Additionally, Barabadi and Naseri [3] reported that in the Artic region or those that have similar operational conditions have the issue of having a lack of performability data that addressed the two performability concepts which include elements such as quality, safety and sustainability. The lack of data would make it difficult to optimize any operational procedure. Therefore, the usage of green solutions would minimize the risk and environmental impacts from incidents such as the spillage of chemicals, thus avoiding or lessening the need for such data. In Kelland [5], several green alternatives developed for the purpose of overcoming the environmental limitation of commercial KHIs were listed. One of the first developed solutions are Antifreeze Proteins (AFP) or Antifreeze Glycoproteins (AFGP) that are derived from fish [12]. The most promising green alternative and readily biodegradable solution that is listed in the Kelland [5] review was *Pectic Polysaccharides* (Pectin). Xu et al. [13] also agreed with Kelland’s assessment. Xu et al. [13] conducted an experiment that compared pectin and a commercial KHI; the PVCap performance concluded that pectin has a better performance and is more economical than its commercial counterpart. Effendi et al. [7] also confirmed that pectin performed better or comparable to PVCap as a KHI. Hence, the usage of these biocomponents would benefit operators in the Artic region or regions with similar conditions.

In the last decade, several biofriendly kinetic hydrate inhibitors were researched to identify their potential, such as polysaccharides (starch, pectin, cellulose and chitosan) and proteins [12,14,15,16]. These studies showed promising results relative to their commercial KHI counterparts, such as Polyvinylpyrrolidone (PVP); yet the mechanism and impact of the molecular structure and functional group composition specifically on amide have not been investigated. Previous research conducted by Effendi et al. [8] investigated the compositional effect of the functional groups in pectin, specifically carboxyl and ester. The study found that the functional group with higher electronegativity performed better compared to the group with lower electronegativity, i.e., carboxyl and ester, respectively.

To understand how pectin is effective as a KHI, the hydrate formation inhibition mechanism should be understood. Based on the review by Bavoh et al. [17] on amino acids, the general inhibition mechanism for pectin could be explained by the hydrogen bond that occurs between water molecules and amino acids, a process called adsorption. For gas hydrates, there are several structures known as Structure I, Structure II and Structure H which defer mainly from the gas that acts as the “guest gas” and the structure arrangement [18]. Similar to the amino acid, pectin would inhibit the formation by adsorption or perturbation (Figure 1). Because it is related to the hydrogen bonding occurring, the chemical used that has high electronegativity would in theory perform better as a KHI.

Pectin’s structure contains a number of functional groups, and the most essential functional groups in pectin’s structure are the hydroxyl, amide and carboxyl groups, which could inhibit the formation of hydrates. Water molecules are attracted to these groups by a polar attraction, preventing them from clumping together. The stronger the hydrogen bond formed between the functional group, the longer the KHI can delay the hydrate formation. This is because a stronger hydrogen bond can potentially create more hydrogen bonds, thus increasing the crystallization energy [19] for hydrate formation. According to Ophardt’s [20] report, the electronegativity of functional groups can be ranked as follows:Amide > Acid (Carboxyl) > Alcohol > Ketone ~ Aldehyde > Amine > Ester > Ether > Alkane

Based on the ranking, it was found that amide functional groups have the highest electronegativity in comparison to other types of functional groups. As a result, there is a higher probability of forming a hydrogen bond with water molecules, which effectively inhibits hydrate formation through adsorption. Low-methoxylated pectin (LMP) has more carboxyl functional groups and high-methoxylated pectin (HMP) has more ester functional groups. These pectin types were classified based on their degree of esterification (DE), with pectin having a DE greater than 50% considered as HMP and those with a DE less than 50% considered as LMP. This determination can be performed by titration [21]. LMPs which have the higher electronegative functional groups were observed to perform better than HMP at certain conditions [7]. Hence, pectin which is a dominantly ester or carboxyl functional group could be amidated to increase the amide functional group presence in the pectin, thus possibly improving the inhibitory performance of pectin. From a patent that manufactures amidated pectin, it suggested that it is safe and economical to use amidated pectin (AMP) as it is also used as a medical ingredient in the pharmaceutical industries and has a property that promotes wound healing [22]. The amidation process as described by Kratchanov et al. [23] would add an amide functional group to the polymer, removing the ester bonds as shown in the following illustration (Figure 2).

The research conducted on the functional group by Yu and Mosbach [24] validates Ophardt’s [20] findings on the functional group where amide is more electronegative than the carboxyl functional group. A variety of hydrogen-bonding complexes having just one hydrogen bond have been examined in the gas phase for amides and carboxylic acids; it was discovered that carboxyl and amide hydrogen bonds are the strongest among all forms of hydrogen-bonding complexes containing only one hydrogen bond. Yu and Mosbach [24] stated that the large difference in the dielectric constant and dipole moments would infer that amide would form stronger hydrogen bonds than the carboxyl functional group. Even so, there still remains a small doubt of the amide and carboxyl functional groups’ relative hydrogen bond strength [25]. A carboxyl functional group forming a hydrogen bond would require higher bond energy compared to amide. Therefore, amide has a higher tendency to form a hydrogen bond comparatively to carboxyl [26,27]. The carboxyl group, with a C=O dipole, is a stronger dipole than the N–C dipole, primarily due to oxygen’s higher electronegativity. Despite this, amides can act as hydrogen bond acceptors because they possess a C=O dipole and, to a lesser extent, a N–C dipole. Furthermore, primary and secondary amides contain N–H dipoles that allow them to act as hydrogen bond donors. Consequently, amides can form hydrogen bonds with water and other protic solvents. The oxygen atom in the C=O group can accept hydrogen bonds from water, while the N–H hydrogen atoms can donate hydrogen bonds. These interactions result in amides having higher water solubility compared to their corresponding hydrocarbons [28]. This research would determine the difference in the functional group effects for inhibiting hydrate formation specifically. For that reason, AMP was evaluated as a KHI.

To evaluate the inhibitors used, Daraboina et al.’s [14] methodology was employed with some consideration from Ke and Kelland’s [29] review. The methodology extract from Daraboina et al. [14] was their improvement from a bulk sample into a sample in capillary tubing which clearly improved the accuracy and hydrate evaluation. Although the use of silica gel was not used in this study due to unavailability, the improvement from a bulk into capillary tubing sample is sufficient to ensure an accurate reading. The consideration from Ke and Kelland [29] was their review on a constant cooling–isobaric evaluation of a hydrate such as the ramping method used in this study which follows the general concept of a hydrate performance evaluation based on the equipment used. For example, a constant cooling method was used in Ajiro et al.’s [30] research where they instead used an autoclave machine that indicated to hydrate formation from the pressure difference, thus pinpointing the hydrate formation temperature. In this study, the ramping method could be conducted with the high-pressure micro-differential scanning calorimeter (HP-µDSC) that involves monitoring the hydrate formation temperature instead of the usual metric, the induction time or pressure changes. This method is much more economical because it further reduces the time taken per trial, depending on its cooling and heating rate (°C/min) and the minimum temperature (°C). Using this constant cooling procedure, it is reportedly less stochastic with less scattered data points than that under constant temperature (isothermal) which is often used when the monitored parameter is the induction time [29]. It is more effective as more runs can also be performed when coupled with Daraboina et al.’s [14] suggestion of using capillary tubing. The device is illustrated in the following figure (Figure 3).

The inhibition performance was evaluated using two different concentrations (0.10 and 1.00 wt.%) of each chemical, Polyvinylpyrrolidone (PVP), AMP, HMP and LMP, as this concentration is within the expected LDHI concentration of 0.1 wt.% to 3.0 wt.% [5] and the range of concentration is sufficient to evaluate the impact of the concentration on the performance of the KHI tested. The experiment was conducted under an isobaric condition with a constant cooling rate to obtain the result as a function of temperature (referred to as the hydrate formation temperature) at a 10MPa pressure. This research attempts to enhance the development of green kinetic hydrate inhibitors, specifically pectin, by relating the hydrate inhibition mechanism to the two or three functional groups (ester, carboxyl and amide). The goal of this research is to quantify the effect of the change in the functional group composition. These chemical modifications (amidation) can potentially improve the inhibition property while maintaining its biodegradability.

The remaining part of this paper proceeds to discuss the methodology used, the results and the analysis of the results obtained and the conclusion of this study. The methodology includes the material used, the method to evaluate the KHI and how the determination is conducted. In the results and discussion, the findings are reported and discussed further based on the tabulated data and the figures illustrated. The conclusion will then summarize the main findings and the way forward for future studies.

## 2. Materials and Methods

### 2.1. Materials

Three types of pectin, namely LMP, HMP and AMP, were acquired from citrus origin by purchasing them through a chemical supplier. LMP has an average degree of esterification (DE) of 25–35%, HMP has an average DE of 72–82% and AMP has a DE of 20–35% and a degree of amidation of 40–55%. In addition, synthetic KHI Polyvinylpyrrolidone (PVP), a commercial KHI, was also obtained. To conduct the experiments, capillary tubes (length 5 mm × ⌀ 2 mm) were acquired, and deionized water was used to prepare the samples, while methane gas was provided in the on-site lab.

### 2.2. Kinetics Hydrate Inhibitor Performance Test

The tested KHIs were evaluated through ramping method as described in Daraboina et al. [31] and Daraboina et al. [14] based on Ke and Kelland’s [29] review. The metric observed for evaluating the KHI performance is the hydrate formation temperature. The conditions were set to an isobaric condition (10 MPa) with a constant cooling rate of −0.2 °C/min until it reached −25 °C, and after 10 min at −25 °C, it would be heated with 0.2 °C/min until it reached ambient. It was expected that at the condition set during experimentation, the tested KHI would not perform well including AMP as the temperature drop until approximately −25 °C which can be considered as a very high subcooling of 37 °C. The lower the hydrate formation temperature recorded, the better the KHI performs as it would require lower hydrate formation temperature to form. So, in an isobaric condition of 10 MPa, the expected formation temperature would be approximately 12.78~13 °C (55 °F or 286.15 °K) based on Sloan Jr. and Koh [32] and McCain [33] hydrate phase diagram.

Figure 4 shows an example of hydrate formation determination where exothermic peak is identified and interpreted to be either ice or hydrate peak. The onset temperature of the exothermic peak is considered as the time or point of detectable hydrate or ice formation. The endothermic peak is used to confirm the presence of ice and hydrate formation occurrence as there is possibility of only one of the formations occurring such as only ice, and it also used for determination of peak as the heat used in exothermic and endothermic should be roughly similar. Thus, if ice endothermic is 100 mW, then the exothermic should be roughly similar. In addition, the peak of either ice and hydrate has certain characteristic that shows that there is either ice or hydrate as described by Maeda et al. [34] where ice formation was typically characterized by a large and sharp exothermic peak, whereas gas hydrate formation was typically characterized by a smaller and more gradual exothermic peak.

Because the samples with a volume of approximately 2–3 µL were placed inside small capillary tubing (with a 2.3 mm diameter and 0.5–1 cm length), more samples can be tested at each trial, evaluating at least 5 samples per trial. In this study, each sample was tested at least 2–3 trials, amounting to 10–15 repetitions per sample. The samples with observable hydrate exothermal peaks were selected and the average values are reported in this work.

All the samples were fresh samples tested; thus, memory effect of each tested KHI was avoided. The samples were isolated from each other as they were placed inside the cell in different capillary tubing, as illustrated in Figure 5 and Figure 6. This number of samples remains to be economical methodologically as it reduces the time taken and increases the accuracy of peak determination. Throughout the experiment, ice peaks were also identified, and hydrate peak could overlap with other identified hydrate and ice peak, as shown in Figure 4. Hence, there is a likelihood of missed hydrate peak. If overlapped with a hydrate peak without clear difference detected, it will be considered as one (1) hydrate peak and similar to one that overlaps with ice peaks. This overlap occurred as there is possibility for the peak to occur at the same time as the sample could form hydrate or ice at the same time as they were separated into different capillary tubing, as described by Daraboina et al. [31]. Water sample without any additives was also tested; this sample would be used as reference to compare between the tested KHI to produce the Relative Inhibitor Performance (*RIP*) [35,36], which is defined as:(1)$$RIP=\frac{X-Y}{Y}$$
where *X* is the hydrate nucleation peak temperature with inhibitor and *Y* is hydrate nucleation peak without inhibitor. Moreover, it should be noted that the test is in a static condition where the performance should be perceived as the upper limit of the tested KHI performance, as real condition involves turbulence and dynamic movement from the environment, such as vibration.

### 2.3. Analysis

A descriptive statistic was used to evaluate the performance of the tested KHI. Based on their hydrate formation temperature, the statistical data helped establish a conclusion on the overall performance. The analysis was conducted using Minitab software. The software develops visuals to properly illustrate the overall performance of the tested KHI. The exothermic peak for both hydrate and ice were determined based on characteristic described from Maeda et al. [34] after determining the hydrate and ice presence from the dissociations (endothermic) peak.

## 3. Results and Discussion

### 3.1. Ramping Method Descriptive Statistics

Table 2 presents an overview of the result obtained from the experimentation conducted. What stands out in the table is that only AMP demonstrates an inhibitory property on average, based on its average nucleation time. The Relative Inhibitor Performance (RIP) was used to summarize the performance of the inhibitors in this investigation, similar to Yaqub et al. [35]. However, instead of using the induction time as the parameter, the hydrate nucleation temperature was used. On average, AMP shows that it has an RIP of 0.07 and a 0.1 improvement as it requires a lower temperature for hydrate to form for the concentrations of 0.10 wt.% and 1.00 wt.%, respectively. These findings confirm that a kinetic evaluation of a gas hydrate does show stochastic behavior similar to previous observations [37,38]. Hence, in the condition set, only AMP was on average able to delay the required hydrate formation temperature. Table 2 is further analyzed descriptively and tabulated in Table 3.

In Table 3, from the mean average of the hydrate formation temperature, a KHI performance ranking was constructed and shown separately by their concentration initially.
0.1 wt.%: AMP > Water >PVP > LMP > HMP
1.0 wt.%: AMP > Water >PVP > LMP > HMP

Although the concentrations of the tested KHIs were different, both concentrations exhibited a similar trend in which AMP outperformed the other KHIs. It is widely acknowledged that the effectiveness of a KHI decreases beyond a certain level of the subcooling temperature. This finding is in line with a previous study conducted by Del Villano et al. [39], which reported that the effective subcooling temperature range was up to 12 °C. Therefore, based on Del Villano et al.’s [39] method, it is assumed that the KHI would lose its effectiveness in the current tested environment once the temperature drops below 1 °C. In the current setup, the solution containing a KHI and water was exposed to subcooling temperatures above 20 °C (−7 °C). Based on the ranking, it appears that, excluding AMP, the other KHIs acted more as promoters than inhibitors. However, if the metric used was the induction time, the findings may have been different.

### 3.2. Kinetic Hydrate Inhibitor Evaluations

From Table 3, the standard deviation estimation indicated that the variation in the hydrate formation temperature for the tested KHI was within the error limits. This indicated that there was a minimal effect of the KHI tested on the hydrate formation. Within the error limits, in general, it seems AMP would be the best KHI to be used as shown in the results obtained for the condition that requires pressure to be maintained (isobaric) at a certain level while the temperature is gradually decreasing (constant cooling). AMP remains to be able to delay the hydrate formation temperature lower than the water hydrate formation temperature despite the high subcooling level. This is further illustrated in Figure 7 as it compared the KHIs tested.

The results, as shown in Figure 7, indicate that AMP has the lowest average hydrate formation temperature of −16.69 °C and −17.12 °C at 0.1 wt.% and 1.0 wt.% concentrations, respectively. This is followed by water, PVP, LMP and HMP for both concentrations. This finding is also in accordance with that of Effendi et al. [7] who demonstrated that PVP, LMP and HMP lose their relative effectiveness at a higher subcooling level and the effect of different concentrations on their inhibitory performance. The KHI performance is within the error margins and is ranked as follows:AMP (1 wt.%) > AMP (0.1 wt.%) > Water > PVP (0.1 wt.%) > PVP (1 wt.%) > LMP (0.1 wt.%) ≥ HMP (0.1 wt.%) > LMP (1 wt.%) > HMP (1 wt.%)

From the ranking of the tested KHIs, excluding AMP, the other KHIs demonstrate that the 0.1 wt.% concentrated sample performs slightly better than the 1.0 wt.% concentrated sample within the tested error limits. At 0.1 wt.%, LMP and HMP have a similar overall performance, with LMP slightly outperforming HMP.

Figure 8 illustrates the Box-Plot graph that presents all of the tested KHIs. Upon closer inspection of both AMP concentrations, their mean values remained below the dashed red line, which represents the mean hydrate formation temperature for water without additives. In contrast, the mean values for the other tested KHIs were on average above the dashed red line. Even so, the tested KHIs still have the potential to delay the hydrate formation temperature. The descriptive analysis concludes that the 0.1 wt.% performs better and more consistent in the results overall, and AMP is concluded to be the best performing KHI during testing in regard to lowering the required hydrate formation temperature.

The descriptive analysis also shows the KHI distribution for each concentration which can be observed from the Box-Plot graph (Figure 8). AMP has the lowest standard deviation (0.1 wt.%) and PVP has the greatest (1.0 wt.%). In comparison to the 0.1 wt.% KHIs, there is a pattern where most of the 1.0 wt.% KHIs have a larger standard deviation. Besides the 0.1 wt.% HMP, the other 0.1 wt.% KHIs including the 1.0 wt.% LMP are relatively more consistent.

It was hypothesized that the AMP would perform relatively better than the LMP and HMP due to the presence of amide. The results of the study indicate that AMP performs better in an isobaric–isothermal environment than LMP and HMP. This is likely due to the higher electronegativity of the amide group, which allows for easier hydrogen bonding with water molecules and inhibits the hydrate formation through adsorption [17]. However, the performance shown by the LMP and HMP at both concentrations suggests that at a lower concentration they were able to disrupt the hydrogen bonding better than their higher concentration counterpart in the environment tested. These results match with Xu et al.’s [13] findings that reported that pectin at 0.25 wt.% performed better than at 0.5 wt.% in inhibiting hydrate formation. Upon comparison of the three pectin types, our findings support the initial hypothesis that a higher electronegative functional group can more effectively inhibit hydrate formation. The observed trend can be attributed to the greater electronegativity of amide relative to carboxyl and carboxyl relative to ester, highlighting the importance of the functional group in determining the performance of pectin as a hydrate inhibitor. These results provide valuable insights into the design of more effective hydrate inhibitors and offer potential avenues for further research in the field.

## 4. Conclusions

In this investigation, the aim was to assess the effectiveness of different pectin types at isobaric–constant cooling conditions and examine the AMP potential improvement over the established LMP and HMP types for hydrate inhibition. This study has identified through the ramping method that AMP performs the best in both tested concentrations in isobaric–constant cooling conditions. This paper has provided a deeper understanding of the functional group in regard to hydrate inhibition. Due to the addition of the amide functional group in pectin, the *pectic polysaccharides* perform well and could overcome the KHI limitation of inhibiting the hydrate at a high subcooling temperature. That being the case, this research has laid the groundwork for future research on AMP, despite its exploratory nature. Perhaps the next natural progression of this study is to conduct experiments at isothermal–isobaric conditions as most of the KHI evaluation was conducted by monitoring the induction time. These results cemented *pectic polysaccharide*’s ability to be an alternative for hydrate inhibition in the cold north or regions that have a similar environment, overcoming the logistical issues and environmental restriction and thus increasing the profitability of developing gas reserves in the cold region.

## Figures and Tables

**Figure 1 polymers-15-02080-f001:**
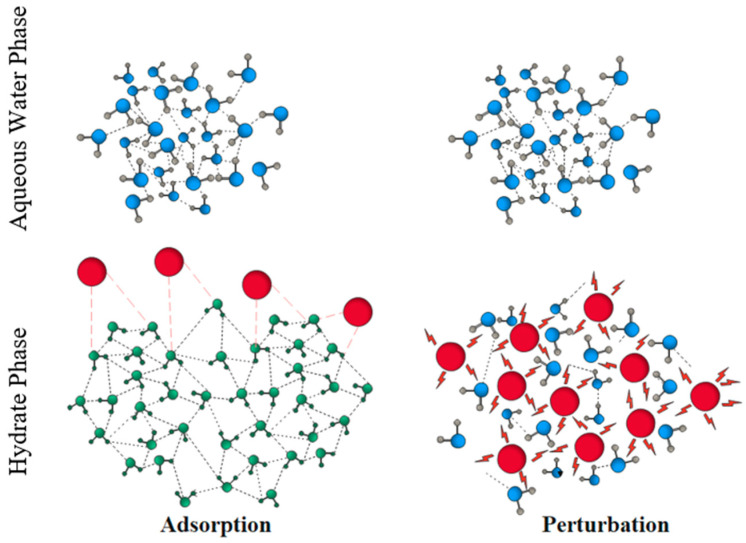
Gas hydrate formation inhibition mechanism: adsorption (**left**) and perturbation (**right**).

**Figure 2 polymers-15-02080-f002:**
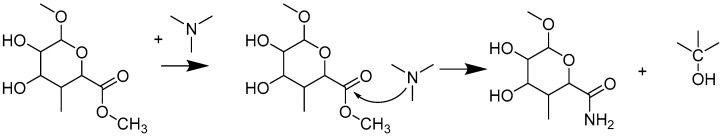
Pectin undergoing amidation.

**Figure 3 polymers-15-02080-f003:**
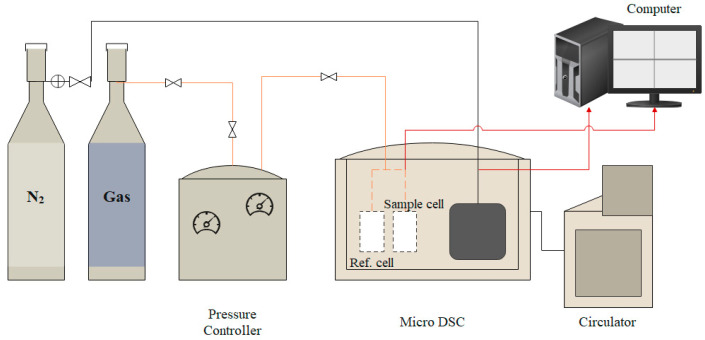
HP Micro-DSC device arrangement.

**Figure 4 polymers-15-02080-f004:**
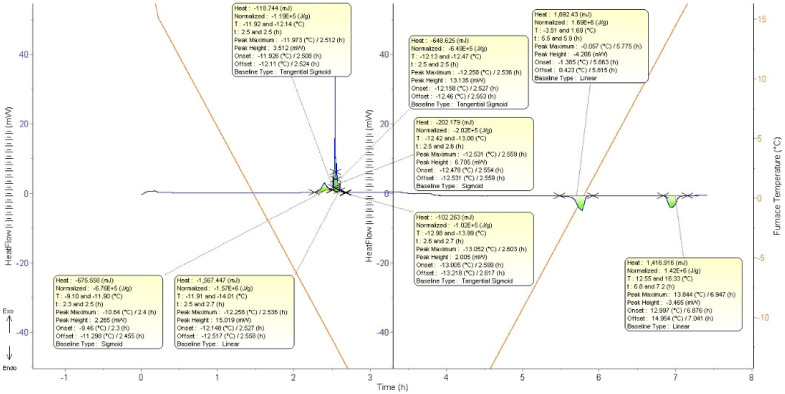
LMP (1.00 wt.%) ramping graph (illustrating the ice and hydrate formation).

**Figure 5 polymers-15-02080-f005:**
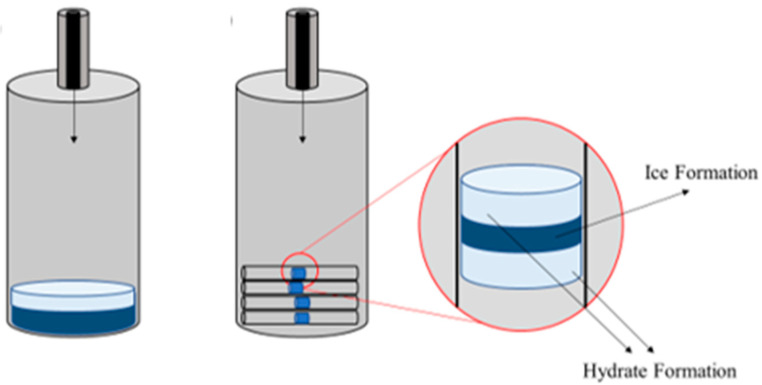
(**Right**) Conventional HP-µDSC sample (bulk). (**Left**) Four open-ended borosilicate tubes are placed horizontally inside the HP-µDSC cell to promote hydrate formation. Adapted from [7].

**Figure 6 polymers-15-02080-f006:**
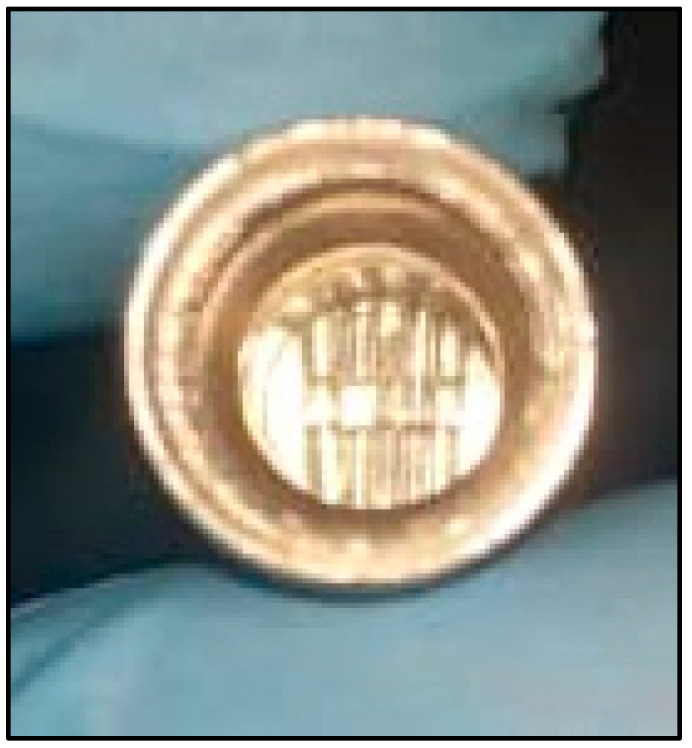
Capillary tubes containing sample placed inside the HP-µDSC cell. Adapted from [7].

**Figure 7 polymers-15-02080-f007:**
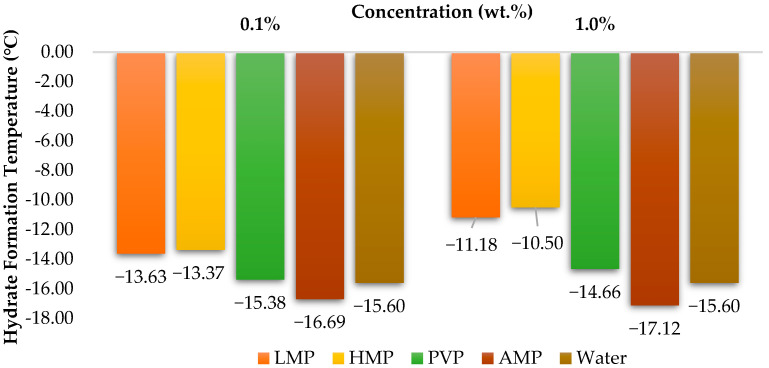
Tested KHIs average hydrate temperature formation.

**Figure 8 polymers-15-02080-f008:**
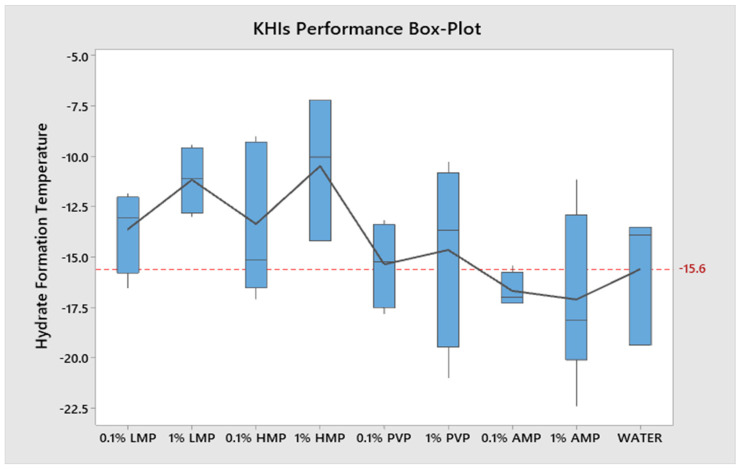
Box-Plot of the tested KHIs performance by concentration.

**Table 1 polymers-15-02080-t001:** Chemical method for hydrate inhibition comparison [11].

Factors	Chemicals		
	KHI	AA	THI
CAPEX/OPEX	Low	Low	High
Volume Required	Low (>1 wt.%)	Low (>1 wt.%)	High (5–60 wt.%))
Toxicity	Low	Low	High
Application	All types of systems	Inapplicable for Gas	All types of systems
Benefits	Tested in gas system	Wide range of subcooling	Predictable models Recoverable (MEG)
Limitations	- High subcooling - Shut-in/cyclic wells	- Restricted to <50 water cut % - Shut-in/cyclic wells	- Logistically difficult - Hazardous

**Table 2 polymers-15-02080-t002:** Tested KHI results from HP-µDSC.

Sample	Concentration (wt.%)	Nucleation Peak at Temperature (°C)						Average Nucleation Peak Temp. (°C)	RIP
		1	2	3	4	5	6		
LMP	0.10%	−13.599	−16.534	−11.877	−12.516	-	-	−13.630	−0.12
	1.00%	−9.460	−13.005	−9.975	−12.294	-	-	−11.184	−0.28
HMP	0.10%	−9.049	−9.565	−15.167	−16.007	−17.085	-	−13.370	−0.14
	1.00%	−7.243	−10.058	−14.199	-	-	-	−10.500	−0.33
PVP	0.10%	−13.206	−13.911	−16.574	−17.815	-	-	−15.380	−0.01
	1.00%	−12.432	−20.994	−10.299	−14.919	-	-	−14.660	−0.06
AMP	0.10%	−15.447	−16.75	−17.247	−17.319	-	-	−16.691	0.07
	1.00%	−13.502	−18.998	−19.335	−22.423	−11.183	−17.251	−17.120	0.10
Water	-	−13.521	−13.917	−19.374	-	-	-	−15.604	-

**Table 3 polymers-15-02080-t003:** Descriptive statistics of the collected data.

Variable (wt.%)	Samples Run	Mean (°C)	Standard Deviation	Minimum (°C)	Interquartile 1 (Q1) (°C)	Median (°C)	Interquartile 3 (Q3) (°C)	Maximum (°C)
0.10% LMP	10	−13.630	2.060	−16.530	−15.800	−13.060	−12.04	−11.88
1.00% LMP	15	−11.184	1.730	−13.005	−12.827	−11.134	−9.589	−9.460
0.10% HMP	15	−13.370	3.780	−17.090	−16.550	−15.170	−9.310	−9.050
1.00% HMP	10	−10.500	3.500	−14.200	−14.200	−10.060	−7.240	−7.240
0.10% PVP	10	−15.380	2.180	−17.820	−17.500	−15.240	−13.380	−13.210
1.00% PVP	15	−14.660	4.620	−20.990	−19.480	−13.680	−10.830	−10.300
0.10% AMP	10	−16.691	0.867	−17.319	−17.301	−16.998	−15.773	−15.447
1.00% AMP	10	−17.120	4.120	−22.420	−20.110	−18.120	−12.920	−11.180
Water	10	−15.600	3.270	−19.370	−19.370	−13.920	−13.520	−13.520

## Data Availability

The data presented in this study are available in the article.

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
