# Peer review of "Amidated Pectic Polysaccharides (Pectin) as Methane Hydrate Inhibitor at Constant Cooling and Isobaric Condition"

_polymers, 2023, doi:10.3390/polym15092080_

Round 1

Reviewer 1 Report

The manuscript studied using different kinds of pectins (low methoxyl, high methoxyl, and amidated) as greener methane hydrate inhibitors to replace the other currently used chemicals. The topic is unique and very interesting too. The work is well-designed and the manuscript is generally well-written in its different parts. I recommend publication after taking the following minor revision in consideration.

-          In the Experimental, please provide the source of different pectins used and also all other specifications, especially for the amidated pectin (degree of amidation,…etc.)

Author Response

Reviewer 1#

Dear Reviewer,

I would like to express my sincere gratitude for taking the time and effort to review my research paper titled "Amidated Pectic Polysaccharides (Pectin) as Methane Hydrate Inhibitor at Constant Cooling and Isobaric Condition". Following are the revision done based on your feedback:

REVIEWER

COMMENTS

REVISION/ REBUTTAL

#1

1.        In the Experimental, please provide the source of different pectins used and also all other specifications, especially for the amidated pectin (degree of amidation,…etc.)

At subsection 2.1, additional information was added as per requested. The details include involves only the DE and Degree of Amidation mainly due to investigating the effect of the different functional group to the inhibitory properties of pectin to methane hydrate.

“Three types of pectin, namely LMP, HMP, and AMP, were acquired from citrus origin. LMP has an average degree of esterification (DE) of 25-35%, HMP has an average DE of 72-82%, and AMP has a DE of 20-35% and a degree of amidation of 40-55%. …” (Refer Line 192-195)

Reviewer 2 Report

Dear Authors thank you very much for this rerearch and this paper, but unfortunately at this point I believe that it needs to be rejected.

There are the main reasons for it

1.       Methodology:

a.       first of all there are just four measurements for each composition. I think that due to stochasticity of hydrate nucleation phenomenon this amount is too low to make any judjements

b.      if you put error bars on the chart shown in figure 5, you‘ll see that the most of the date are within the experimental errors, which probably mean that under your experimental conditions there is no difference between the studied systems in terms of hydrate nucleation

c.       There is significant lack in the description of experimental procedure: there I no information on whether the sample was changed after each experiment or not, if not, how you got rig of memory effect etc. Also if you placed several containers to the cell simultaneously to save time, how you isolated them to avoid cross-nucleation (when crystals of the sample with nucleated hydrate touch non-nucleated hydrate)

d.      Performance of inhibitor is described with parameter “% of T”… I may be wrong but I have never seen that nucleation delay was measured with this unit. If I am wrong, please provide some paper from respectable group in which this parameter was used.

e.      Which mass of water of solution was loaded into the cell each time? Was it kept constant from experiment to experiment?

f.        “High” and “low” treated pectin : needs to be quantified.

2.       Introduction:

a.       From the information you provided in intro, I can understand that your research is rather focused on solution which could be applied in North Sea. Please provide more specific information about it, do not give (Arctic, -30oC etc. in arctic, typically there are subcooulings at which KHI do not work) it as general information for all places, because in this way is turns to be incorrect. This is high impact journaL (IF~5), so one could expect very careful treatment of background information, so please double check every fact.

3.       Language style:

a.       I understand that you are not a native speaked of english. But anyway you need to make you sentences clear and avoid grammar mistakes. In present time there are number of tools in internet allowing the correction of the language style.  Several of them might be free of charge.

Lines 8-10: methanol have been successfully applied for prevention of hydrate formation in pipelines in northern oilfields in Alaska, Canada and Russia for many years as the most robust prevention strategy even at tempatures lower than -30oC.

Line 11: delays hydrate nucleation?

Line 16: what does “electroneganive” mean in this context? Commonly this term is applied to evaluate ability of atoms to donate or accept electrons

Line 17-18: what is low and high Methoxylated? Is there ane quantitative measure? For an example number of functional groups per number of monomers?

Line 19: how did you apply percentage of tempearture? May be I am wrong but I’ve never seen this approach. Please provide how you calculated this value and reason of using this term over more traditionally used function of survival vs temperature.

Line 32: “uncertain harsh weather conditions” - ?

Line 37: there can be much colder temperatures in arctic regions, down to -50°C and even colder.

Line 44: are these alcohols biodegradable at all? Please provide references for studies where biodegratation of these alcohols was studied.

Line 44: companies operating in North Sea have stricter safety regulations than anywhere else. But up to my knowledge glycols are still used there for hydrate prevention. 

Table 1: could you please clarify in text somewhere how KHI’s can be dosed in gas pipelines (if it is production pipeline which can contain water, AA’s can be applied too I guess) or put reference. Also pleae put a reference for not aplicability of AA’s for shut-in. I believe that flow can be easily restarted with suspension of hydrates. I saw number of SPE papers on it.

Line 55: the meaning of the sentence is not clear

Line 61: Malcolm is a man but not a woman. At least he was a man when I saw him last time. https://www.uis.no/en/research/green-and-sustainable-chemistry

Line 78: was it natural pectine or chemically treated?

Line 82: I belive that last time I saw a lot of papers showing potential of amino-acids for promotion of hydrates formation. I refer you to work of the group led by professor Linga. Aslo in this form the sentence does not make any sence.

Figure 1: please place the reference for the picture

Line 97: should there be a ref or it is your assumption?

Line 98; ranked in term of performance for hydrate inhibition?

Line 143: is there a trade-off between cooling rate and precision of data?

Line 175: it is not clear wheter water or khi solution was reloaded after each experiment. Also, was your capillary tube completely filled with water?

Please describe the experimental procedure in more details. Do you have DSC that allows you to perform the experiments with 5 samples simultaneosly? Or you somehow placed 5 tubes into one cell? May be a picture would be helpful to understand it.

Line 233: what the term “%” corresponds to?

Table 2 in this form it is not possible to understand which data belong to which experiment. Please make the table clear and inambigous. Put STD to mean values.

General comment to experimental work. 4 experiments is not enough to judge about performance of KHI. Nucleation process is stochastic. Commonly approach of analysis if KHI involves biulding of “survival function”: time vs percentage of experiments where nucleation did not happen up to this time as well. I can refer you to the paper 10.1016/j.egypro.2019.01.530

Table 3: what is Q1, what is Q3?

Figure 5 if you put error bars on the chart, probably there will be overlapping of the nucleation time data for the most of the experiments (at least I can gues it looking at the results presented in figure 5). Essentially it means that there in no differnce of nucleation temperature in your experiments for all tested systems and you need to redesign the experimental procedure (slow down the cooling rate, change the size of the sample etc.). Also please provide the legend for black and yellow bars.

Figure 6 : what black and red dash lines on the picture are corresponding to?

Author Response

Reviewer 2#

I would like to extend my sincerest gratitude to you for taking the time to review my research paper titled "Amidated Pectic Polysaccharides (Pectin) as Methane Hydrate Inhibitor at Constant Cooling and Isobaric Condition". Your insightful feedback and thoughtful comments have been instrumental in improving the quality of my work. I am grateful for your expertise and attention to detail, which have been invaluable in making my paper more impactful and relevant. Thank you once again for your time and effort in providing me with such valuable feedback. Following are the revision done based on your feedback:

REVIEWER

COMMENTS

REVISION/ REBUTTAL

#2

Methodology:

1)     first of all there are just four measurements for each composition. I think that due to stochasticity of hydrate nucleation phenomenon this amount is too low to make any judgments.

After conducting four measurements with a sample total of 10-15 samples, each composition was further screened, resulting fewer usable results representing the hydrate formation peak. This approach is consistent with previous studies conducted by Lachance, Sloan, and Koh (2009), Maeda, Kelland, and Wood (2018), Daraboina, Malmos Perfeldt, and von Solms (2015), and Daraboina, Ripmeester, Walker, and Englezos (2011), who also used DSC with fewer results for each composition. While increasing the number of samples may provide better interpretation, the current results are deemed sufficient, as supported by previous research. Although I agree, increase number of samples would allow better interpretation but similar to previous researchers, current results should be sufficient. This was clarified in the following line:

“In this study, each sample were tested at least 2-3 trials, amounting to 10 – 15 repetition per samples. The samples with observable hydrate exothermal peaks were selected and the average value were reported in this work.” (Refer Line 230 – 233)

The table 3 were also updated to reflect the amount of sample tested under the sample runs column.

1.      if you put error bars on the chart shown in figure 5, you‘ll see that the most of the date are within the experimental errors, which probably mean that under your experimental conditions there is no difference between the studied systems in terms of hydrate nucleation

I have revised the statements that address the differences between the tested KHI. While it is true that there is no significant difference between the tested KHIs, the small differences that do exist are within the acceptable error limits, indicating a minimal effect.

“From table 3, the standard deviation estimation indicated that the variation in the hydrate formation temperature for the tested KHI were within in the error limits. This indicated that there was a minimal effect of the KHI tested to the hydrate formation. Within the error limits in general …” (Refer Line 303 – 306)

However, comparing within these error margins does show an observable difference between the tested KHIs. Based on their mean values, the tested KHIs can be ranked accordingly.

2.      There is significant lack in the description of experimental procedure: there I no information on whether the sample was changed after each experiment or not, if not, how you got rig of memory effect etc. Also if you placed several containers to the cell simultaneously to save time, how you isolated them to avoid cross-nucleation (when crystals of the sample with nucleated hydrate touch non-nucleated hydrate)

More details were added to address the memory effect concern:

“All the samples were fresh samples tested thus memory effect of each tested KHI was avoided.” (Refer Line 234 – 235)

“… samples with a volume of approximately (2-3 µL) … tubing (with a 2.3mm diameter and 0.5-1 cm length), …” (Refer Line 228 - 229)

The isolation of each sample was also addressed and figure were added to show how the sample were placed.

“The sample was isolated from each other as there are placed inside the cell in different capillary tubing as illustrate in Figure 5 and Figure 6.” (Refer Line 235– 236)

3.      Performance of inhibitor is described with parameter “% of T”… I may be wrong but I have never seen that nucleation delay was measured with this unit. If I am wrong, please provide some paper from respectable group in which this parameter was used.

The performance parameter remains to be the average hydrate nucleation temperature peak, however, the “% of T” or difference to water are simply to make it easier for reader to immediately see the differences relative to deionized water without hydrate inhibitors which is similar to Relative Inhibitor Performance (RIP) from Yaqub, Lal, Partoon, and Mellon (2018). Therefore, RIP was added in Table 2. In term of the experimental run, I have conducted in actuality 10-15 runs per sample.

4.      Which mass of water of solution was loaded into the cell each time? Was it kept constant from experiment to experiment?

The volume of each sample was kept at similar volume and are kept constant in term of injected volume for each run for all compositions.

“… samples with a volume of approximately (2-3 µL) … tubing (with a 2.3mm diameter and 0.5-1 cm length), …” (Refer Line 228 - 229)

5.      High” and “low” treated pectin: needs to be quantified.

Assuming that the terms "high" and "low" refer to the determination of whether a pectin is classified as HMP or LMP, further elaboration was included to distinguish the differences between the two.

“These pectin types were classified based on their Degree of Esterification (DE), with pectin having a DE greater than 50% considered as HMP and those with a DE less than 50% considered as LMP.” (Refer Line 112 – 114)

Introduction:

From the information you provided in intro, I can understand that your research is rather focused on solution which could be applied in North Sea. Please provide more specific information about it, do not give (Arctic, -30oC etc. in arctic, typically there are subcooling at which KHI do not work) it as general information for all places, because in this way is turns to be incorrect. This is high impact journal (IF~5), so one could expect very careful treatment of background information, so please double check every fact.

The purpose of the introduction is to provide context for the frequent hydrate issues in the studied region. Given the cool environment which can drop to -30℃, it is expected that gas fields in this region would face hydrate issues. The KHI solution may not be effective in such conditions.

Therefore, the evaluation was based on hydrate formation temperature to identify the extent to which the KHI solution can delay hydrate formation in terms of temperature. To clarify, the example of the new gas field was used to highlight the consideration required when operating a gas field in that region.

The statement and abstract have been revised to improve the clarity. The changes made were intended to enhance the delivery of the research content, and to address any areas of confusion or ambiguity that may have been present in the original text. Thank you for your feedback.

Language style:

I understand that you are not a native speaked of english. But anyway, you need to make you sentences clear and avoid grammar mistakes. In present time there are number of tools in internet allowing the correction of the language style.  Several of them might be free of charge.

1.      Lines 8-10: methanol have been successfully applied for prevention of hydrate formation in pipelines in northern oilfields in Alaska, Canada and Russia for many years as the most robust prevention strategy even at temperatures lower than -30oC.

While it is true that methanol can be used as a hydrate preventive measure, it should be noted that this chemical is known to be hazardous and can pose harm to the environment. Furthermore, it requires special treatment prior to disposal. In addition, the volume of methanol required to effectively prevent hydrate formation can be significant. This is where KHI has an advantage, as it can achieve similar results with a lower dosage, making it a more sustainable and efficient solution.

2.      Line 11: delays hydrate nucleation?

While hydrate nucleation and hydrate formation are distinct in terms of their respective processes, it is still true that delaying nucleation will also delay formation. However, it should be noted that the term 'hydrate formation' was used in this study based on the observed parameter of 'hydrate formation temperature'.

3.      Line 16: what does “electronegative” mean in this context? Commonly this term is applied to evaluate ability of atoms to donate or accept electrons

The electronegative meant that a higher electronegative molecule has a higher likelihood to form hydrogen bonding. This will inhibit hydrate formation through adsorption.

“Yu and Mosbach [24] stated that the large difference in dielectric constant and dipole moments would infer that amide would form stronger hydrogen bonds than carboxyl functional group.” (Refer Line 132 - 134)

“The carboxyl group, with a C=O dipole, is a stronger dipole than the N–C dipole, primarily due to oxygen's higher electronegativity. Despite this, amides can act as hydro-gen bond acceptors because they possess a C=O dipole and, to a lesser extent, a N–C dipole. Furthermore, primary and secondary amides contain N–H dipoles that allow them to act as hydrogen bond donors.” (Refer 138 – 142)

4.     Line 17-18: what is low and high Methoxylated? Is there ane quantitative measure? For an example number of functional groups per number of monomers?

The details were added

“These pectin types were classified based on their Degree of Esterification (DE), with pectin having a DE greater than 50% considered as HMP and those with a DE less than 50% considered as LMP.” (Refer Line 112 – 114)

The DE of both were stated from the chemical supplier.

5.      Line 19: how did you apply percentage of tempearture? Maybe I am wrong but I’ve never seen this approach. Please provide how you calculated this value and reason of using this term over more traditionally used function of survival vs temperature.

The percentage of temperature were used similarly to RIP (Yaqub et al., 2018). Therefore, amendment was done to used RIP as a measure to evaluate the performance of the tested KHI.

6.      Line 32: “uncertain harsh weather conditions” - ?

Just to state that if utilized common hydrate chemical solution such as MeOH, it requires high volume to prevent hydrate formation for sudden downtime.

7.      Line 37: there can be much colder temperatures in arctic regions, down to -50°C and even colder.

Yes, it’s true that the temperature could drop even below -50℃, but the sentence used just to give context to reader that the environment can be at an extreme cold.

8.      Line 44: are these alcohols biodegradable at all? Please provide references for studies where biodegratation of these alcohols was studied.

The statement has been revised to provide accurate information. It was initially written incorrectly, suggesting that MEG and MeOH are both toxic and have poor biodegradation. The example used to illustrate relatively toxic and poorly biodegradable hydrate inhibitors should instead be commercial hydrate inhibitor such as commercial KHI.

“However due to the environmental restriction, it could make it difficult to use common commercial hydrate inhibitor due to its toxicity or/and poor biodegradation (Kelland, 2018; Paz & Netto, 2020).” (Refer Line 38 – 40)

9.      Line 44: companies operating in North Sea have stricter safety regulations than anywhere else. But up to my knowledge glycols are still used there for hydrate prevention. 

The use of MEG as a THI is still common and reliable, as long as it is available. However, due to its volume requirements, it can pose logistical challenges for operators to maintain. LDHI can help address this issue since it can be used at lower dosages.

10.   Table 1: could you please clarify in text somewhere how KHI’s can be dosed in gas pipelines (if it is production pipeline which can contain water, AA’s can be applied too I guess) or put reference. Also please put a reference for not applicability of AA’s for shut-in. I believe that flow can be easily restarted with suspension of hydrates. I saw number of SPE papers on it.

The reference for Table 1 has been added, which tabulated the limitations and benefits of THI, KHI, and AA. While AA can be used as long as there is available oil and a low water cut, it still has limitations compared to THI as both LDHI types are still time dependent. A successful injection procedure for KHI was also added:

“Furthermore, Lavallie, et al. [11] have reported successful applications of KHI in the DOLPHIN field in Qatar which is a sour gas/condensate field and have also discussed their injection procedure. They utilized continuous injection with the first injection being performed offshore to mitigate the risk of hydrate formation, and the second injection was carried out onshore as required at the inlet [11].” (Refer Line 45 – 50)

11.   Line 55: the meaning of the sentence is not clear

The sentence is stating that the region's limited historical data poses a challenge for operators to effectively optimize procedures, including those related to environmental aspects. However, utilizing natural and green solutions could potentially reduce concerns such as spillage and overall make the operator's job easier.

12.   Line 61: Malcolm is a man but not a woman. At least he was a man when I saw him last time. https://www.uis.no/en/research/green-and-sustainable-chemistry

Thank you for the correction, the sentence has been corrected.

“In Kelland [5], several green alternatives developed for the purpose of overcoming the environmental limitation of commercial KHI was listed.” (Refer Line 61 – 62)

13.   Line 78: was it natural pectine or chemically treated?

It is the pectin that was investigated in Effendi, et al. [8], the sentence was re-written to make it clearer.

“Previous research conducted by Effendi et al. [8] investigated the compositional effect of functional groups in pectin, specifically carboxyl and ester. The study found that the functional group with higher electronegativity performed better compared to the group with lower electronegativity, i.e., carboxyl and ester, respectively” (Refer Line 77 – 80)

14.   Line 82: I belive that last time I saw a lot of papers showing potential of amino-acids for promotion of hydrates formation. I refer you to work of the group led by professor Linga. Aslo in this form the sentence does not make any sence.

The statement was re-written as it mainly using the Bavoh, et al. [17] research as reference on the mechanism of hydrate inhibition.

“Based on the review by Bavoh, et al. [17] on amino acids, the general inhibition mechanism for pectin could be explained by the hydrogen bond that occurs between water molecules and amino acids, a process called adsorption. (Refer Line 82 – 84)

15.   Figure 1: please place the reference for the picture

To clarify, Figure 1 was created using EDraw Max and was not extracted from any other research paper.

16.   Line 97: should there be a ref or it is your assumption?

It is an assumption since adsorption depends on the amount of effective hydrogen bonds form which increase the required crystallization energy to form hydrate.

“The stronger the hydrogen bond formed between the functional group, the longer the KHI can delay the hydrate formation. This is because a stronger hydrogen bond can potentially create more hydrogen bonds, thus increasing the crystallization energy (Guan, 2010) for hydrate formation.” (Refer Line 96 – 99)

17.   Line 98; ranked in term of performance for hydrate inhibition?

The sentence was clarified; the functional group were ranked based on their electronegativity

“According to Ophardt (2003) reports, the electronegativity of functional groups can be ranked as follows:” (Refer Line 99 – 100)

18.   Line 143: is there a trade-off between cooling rate and precision of data?

The study was conducted using the lowest possible cooling rate (0.2 ℃/min) that the HPµDSC equipment could reliably achieve, as a slower cooling rate allows for greater precision in the data by providing sufficient time for the detection of the nucleation point.

19.   Line 175: it is not clear wheter water or khi solution was reloaded after each experiment. Also, was your capillary tube completely filled with water?

The volume of each sample was kept at similar volume and are kept constant in term of injected volume for each run for all compositions. Each trial used fresh sample from each composition.

“All the samples were fresh samples tested thus memory effect of each tested KHI was avoided.”

“… samples with a volume of approximately (2-3 µL) … tubing (with a 2.3mm diameter and 0.5-1 cm length), … (Refer Line 227 – 228)

The tubing was not completely filled as shown in Figure 5 and 6. (Refer Line 254 and 257)

20.   Please describe the experimental procedure in more details. Do you have DSC that allows you to perform the experiments with 5 samples simultaneously? Or you somehow placed 5 tubes into one cell? May be a picture would be helpful to understand it.

Yes, each trial has at least 5 samples. It is as shown in Figure 5 and 6.

21.   Line 233: what the term “%” corresponds to?

I assume it is regarding the AMP, initially the percentage meant that AMP manage to delayed the hydrate formation temperature relative to non-additive deionized water. Showing a slight improvement of 6.97% and 9.72% if used AMP with a concentration of 0.1wt.% and 1.0 wt.%, respectively. However, this was changed by using RIP.

22.   Table 2 in this form it is not possible to understand which data belong to which experiment. Please make the table clear and inambigous. Put STD to mean values.

The data in Table 2 is showing the accepted samples for analyzing as their exothermic peak were able to be observed. This table is used to make the reader easier to identify the tested KHI performance; in term of RIP. The table 2 were also rectify to make it visually clearer. While the STD and means are tabulated in Table 3.

23.   General comment to experimental work. 4 experiments is not enough to judge about performance of KHI. Nucleation process is stochastic. Commonly approach of analysis if KHI involves biulding of “survival function”: time vs percentage of experiments where nucleation did not happen up to this time as well. I can refer you to the paper 10.1016/j.egypro.2019.01.530

After conducting four measurements with a sample total of 10-15 samples, each composition was further screened, resulting fewer usable results representing the hydrate formation peak. This approach is consistent with previous studies conducted by Lachance, Sloan, and Koh (2009), Maeda, Kelland, and Wood (2018), Daraboina, Malmos Perfeldt, and von Solms (2015), and Daraboina, Ripmeester, Walker, and Englezos (2011), who also used DSC with fewer results for each composition. While increasing the number of samples may provide better interpretation, the current results are deemed sufficient, as supported by previous research. Although I agree, increase number of samples would allow better interpretation but similar to previous researchers, current results should be sufficient. This was clarified in the following line:

“In this study, each sample were tested at least 2-3 trials, amounting to 10 – 15 repetition per samples. The samples with observable hydrate exothermal peaks were selected and the average value were reported in this work.” (Refer Line 229 – 232)

The table 3 were also updated to reflect the amount of sample tested under the sample runs column. The study also has included RIP to compare the performance.

24.   Table 3: what is Q1, what is Q3?

The Q1 and Q3 represent the interquartile 1 and 3.

25.   Figure 5 if you put error bars on the chart, probably there will be overlapping of the nucleation time data for the most of the experiments (at least I can gues it looking at the results presented in figure 5). Essentially it means that there in no differnce of nucleation temperature in your experiments for all tested systems and you need to redesign the experimental procedure (slow down the cooling rate, change the size of the sample etc.). Also please provide the legend for black and yellow bars.

I have revised the statements that address the differences between the tested KHI. While it is true that there is no significant difference between the tested KHIs, the small differences that do exist are within the acceptable error limits, indicating a minimal effect.

“From table 3, the standard deviation estimation indicated that the variation in the hydrate formation temperature for the tested KHI were within in the error limits. This indicated that there was a minimal effect of the KHI tested to the hydrate formation. Within the error limits in general …” (Refer Line 302 – 305)

However, comparing within these error margins does show an observable difference between the tested KHIs. Based on their mean values, the tested KHIs can be ranked accordingly. The figure 7 were also updated including the missed legend for the other bars.

26.   Figure 6 : what black and red dash lines on the picture are corresponding to?

It is the mean of water without additives to form hydrate formation temperature.

Reviewer 3 Report

In this study, the authors reported amidated pectic polysaccharides (pectin) as methane hydrate inhibitor at constant cooling and isobaric condition, and investigated hydrate formation as the tested samples using HP μ-DSC. The effect of the functional  groups in pectin on hydrate inhibition was discussed. The manuscript is very well-organized, the experimental work is well conducted. I would like to recommend this manuscript for publication after minor revision.

Some comments to the Authors:

1. By comparison of LMP, HMP, and AMP, the corresponding inhibition mechanism need be discussed in detail. 2. The effect of average degree of esterification for AMP on hydrate inhibition need be discussed. 3. Some style need be checked. For example: (1) In line 176, “2.1……..” should be revised to “2.2…….”. (2) In line 221, “2.1……..” should be revised to “2.3…….”.

Author Response

Reviewer 3#

I am writing to express my sincere appreciation for taking the time and effort to review my research paper titled "Amidated Pectic Polysaccharides (Pectin) as Methane Hydrate Inhibitor at Constant Cooling and Isobaric Condition". I am truly grateful for your time and expertise in analyzing my work. Following are the revision done:

REVIEWER

COMMENTS

REVISION/ REBUTTAL

#3

Some comments to the Authors:

By comparison of LMP, HMP, and AMP, the corresponding inhibition mechanism need be discussed in detail.

Additional information/paragraph were added to further elaborate the comparison between the pectin types.

“It was hypothesized that AMP would perform relative better than LMP and HMP due to the presence of amide. The results of the study indicate that AMP perform better in an isobaric – isothermal environment than LMP and HMP. This is likely due to the higher electronegativity of the amide group, which allows for easier hydrogen bonding with water molecules and inhibits hydrate formation through adsorption [15]. How-ever, the performance shown by LMP and HMP at both concentrations suggest that at lower concentration, they were able to disrupt the hydrogen bonding better than their higher concentration counterpart in the environment tested. These results match with Xu, et al. [11] findings that reported that pectin at 0.25 wt.% perform better than 0.5 wt.% in inhibiting hydrate formation. Upon comparison of the three pectin types, our findings support the initial hypothesis that a higher electronegative functional group can more effectively inhibit hydrate formation. The observed trend can be attributed to the greater electronegativity of amide relative to carboxyl and carboxyl relative to ester, highlighting the importance of the functional group in determining the performance of pectin as a hydrate inhibitor. These results provide valuable insights into the design of more effective hydrate inhibitors and offer potential avenues for further re-search in the field.” (Refer Line 347 – 362)

The effect of average degree of esterification for AMP on hydrate inhibition need be discussed. 

The DE of the purchased amidated pectin, which ranges from approximately 20-35%, was not discussed in this study. Each of the pectin types contains carboxyl, ester, and amide functional groups, but the dominant functional group determines the pectin type. Due to limitations, specifically observing the effect of DE on AMP was challenging. Therefore, this study focused on comparing pectin types that are dominantly composed of their respective functional groups for hydrate inhibition.

Some style need be checked. For example: (1) In line 176, “2.1……..” should be revised to “2.2…….”. (2) In line 221, “2.1……..” should be revised to “2.3…….”.

The line was revised and amended based on the Journal formatting. Thank you for pointing the error.

Round 2

Reviewer 2 Report

Dear authors thank you very much for the correction of the paper; from my point of view not it looks much better.  But again mainly due to the first point I’ll leave it to editor whether to publish it or not.

Best regards

1.      However, comparing within these error margins does show an observable difference between the tested KHIs. Based on their mean values, the tested KHIs can be ranked accordingly. The figure 7 were also updated including the missed legend for the other bars.

a.       I can’t agree with that, if all results are within the experimental error this this means that they are pretty much the same and you need to change something in your experimental procedure or to revise your conclusions dramatically.

2.       In this study, each sample were tested at least 2-3 trials, amounting to 10 – 15 repetition per samples. The samples with observable hydrate exothermal peaks were selected and the average value were reported in this work

a.       If hydrate did not nucleate in the holding period, probably this means that you just did not wait enough for it. This approach may distort the statistics of nucleation.

3.       All the samples were fresh samples tested thus memory effect of each tested KHI was avoided.” (Refer Line 234 – 230)

a.       in previous statement you are writing that “each sample was tested at least 2-3 trials”. It is still not clear if you tried to get rid of memory effect between the trials with each sample.

4.       The performance parameter remains to be the average hydrate nucleation temperature peak, however, the “% of T” or difference to water are simply to make it easier for reader to immediately see the differences relative to deionized water without hydrate inhibitors which is similar to Relative Inhibitor Performance (RIP) from Yaqub, Lal, Partoon, and Mellon (2018). Therefore, RIP was added in Table 2. In term of the experimental run, I have conducted in actuality 10-15 runs per sample.

a.       I still can’t see the reason to you “RIP” values to describe the performance of KHI. You are using oC, some people use K, Americans use Farengheits. Which “RIP” value has to be used? Also please use the standard SI measurement units to describe your results.

5.       While hydrate nucleation and hydrate formation are distinct in terms of their respective processes, it is still true that delaying nucleation will also delay formation. However, it should be noted that the term 'hydrate formation' was used in this study based on the observed parameter of 'hydrate formation temperature'.

a.       Please avoid the usage of “hydrate formation” instead of hydrate nucleation. The term “Hydrate formation temperature  ” usually is used to describe the stability of hydrates at certain pressure on P-T phase diagram.

6.       The electronegative meant that a higher electronegative molecule has a higher likelihood to form hydrogen bonding. This will inhibit hydrate formation through adsorption

a.       the term “electronegative ” cannot be used to describe the ability to form hydrogen bonds. Oxygen atom in more electronegative but not the amide group.

7.      The reference for Table 1 has been added, which tabulated the limitations and benefits of THI, KHI, and AA. While AA can be used as long as there is available oil and a low water cut, it still has limitations compared to THI as both LDHI types are still time dependent. A successful injection procedure for KHI was also added:

8.      “Furthermore, Lavallie, et al. [11] have reported successful applications of KHI in the DOLPHIN field in Qatar which is a sour gas/condensate field and have also discussed their injection procedure. They utilized continuous injection with the first injection being performed offshore to mitigate the risk of hydrate formation, and the second injection was carried out onshore as required at the inlet [11].” (Refer Line 45 – 50)

a.       sourgas / condensate in not gas system

9.      I have revised the statements that address the differences between the tested KHI. While it is true that there is no significant difference between the tested KHIs, the small differences that do exist are within the acceptable error limits, indicating a minimal effect.

10.  “From table 3, the standard deviation estimation indicated that the variation in the hydrate formation temperature for the tested KHI were within in the error limits. This indicated that there was a minimal effect of the KHI tested to the hydrate formation. Within the error limits in general …” (Refer Line 302 – 305)